# Practical Schemes for Finding Near-Stationary Points of Convex Finite-Sums

## Abstract

The problem of finding near-stationary points in convex optimization has not been adequately studied yet, unlike other optimality measures such as the function value. Even in the deterministic case, the optimal method (OGM-G, due to Kim and Fessler [33]) has just been discovered recently. In this work, we conduct a systematic study of algorithmic techniques for finding near-stationary points of convex finite-sums. Our main contributions are several algorithmic discoveries: (1) we discover a memory-saving variant of OGM-G based on the performance estimation problem approach [19]; (2) we design a new accelerated SVRG variant that can simultaneously achieve fast rates for minimizing both the gradient norm and function value; (3) we propose an adaptively regularized accelerated SVRG variant, which does not require the knowledge of some unknown initial constants and achieves near-optimal complexities. We put an emphasis on the simplicity and practicality of the new schemes, which could facilitate future developments.

## 1 Introduction

Classic convex optimization usually focuses on providing guarantees for minimizing function value. For this task, the optimal (up to constant factors) Nesterov's accelerated gradient method (NAG) [40, 41] has been known for decades, and there are even methods that can exactly match the lower complexity bounds [30, 17, 55, 18]. On the other hand, in general non-convex optimization, near-stationarity is the typical optimality measure, and there has been a flurry of recent research devoted to this topic [25, 26, 23, 28, 21, 60]. Recently, there has been growing interest on devising fast schemes for finding near-stationary points in convex optimization [42, 2, 22, 7, 31, 32, 33, 27, 15, 14]. This line of research is basically driven by the following facts.

- Nesterov [42] studied the problem with a linear constraint: $f(x^\star) = \min_{x \in Q} \{f(x) : Ax = b\}$, where $Q$ is a convex set and $f$ is strongly convex. Assuming that $Q$ and $f$ are simple, we can focus on the dual problem $\phi(y^\star) = \max_y \{\phi(y) \triangleq \min_{x \in Q} \{f(x) + \langle y, b - Ax \rangle\}\}$. Clearly, the dual objective $-\phi(y)$ is smooth convex. Letting $x_y$ be the unique solution to the inner problem, we have $\nabla \phi(y) = b - Ax_y$. Note that $f(x_y) - f(x^\star) = \phi(y) - \langle y, \nabla \phi(y) \rangle - \phi(y^\star) \le \|y\| \|\nabla \phi(y)\|$. Thus, in this problem, the quantity $\|\nabla \phi(y)\|$ serves as a measure of both primal optimality $f(x_y) - f(x^\star)$ and feasibility $\|b - Ax_y\|$, which is better than just measuring the function value.

- Matrix scaling [50] is a convex problem and its goal is to find near-stationary points [4, 9].

- Gradient norm is readily available, unlike other optimality measures ($f(x) - f(x^\star)$ and $\|x - x^\star\|$), and is thus usable as a stopping criterion. This fact motivates the design of several parameter-free algorithms [43, 39, 27], and their guarantees are established on the gradient norm.

- Designing schemes for minimizing the gradient norm can inspire new non-convex optimization methods. For example, SARAH [46] was designed for convex finite-sums with gradient-norm measure, but was later discovered to be the near-optimal method for non-convex finite-sums [21, 47].

Table 1: Finding near-stationary points $\|\nabla f(x)\| \leq \epsilon$ of convex finite-sums.

| | Algorithm | | Complexity | Remark |
|---|---|---|---|---|
| **IFC** | GD | [33] | $O(\frac{n}{\epsilon^2})$ | |
| | Regularized NAG* | [7] | $O(\frac{n}{\epsilon} \log \frac{1}{\epsilon})$ | |
| | OGM-G | [33] | $O(\frac{n}{\epsilon})$ | $O(\frac{1}{\epsilon} + d)$ memory, optimal in $\epsilon$ |
| | M-OGM-G | [Section 3.1] | $O(\frac{n}{\epsilon})$ | $O(d)$ memory, optimal in $\epsilon$ |
| | L2S | [37] | $O(n + \frac{\sqrt{n}}{\epsilon^2})$ | Loopless variant of SARAH [46] |
| | Regularized Katyusha* | [2] | $O((n + \frac{\sqrt{n}}{\epsilon}) \log \frac{1}{\epsilon})$ | Requires the knowledge of $\Delta_0$ |
| | R-Acc-SVRG-G* | [Section 5] | $O((n \log \frac{1}{\epsilon} + \frac{\sqrt{n}}{\epsilon}) \log \frac{1}{\epsilon})$ | Without the knowledge of $\Delta_0$ |
| **IDC** | GD | [42, 54] | $O(\frac{n}{\epsilon})$ | |
| | NAG / NAG + GD | [32] / [42] | $O(\frac{n}{\epsilon^{2/3}})$ | |
| | Regularized NAG* | [42, 27] | $O(\frac{n}{\sqrt{\epsilon}} \log \frac{1}{\epsilon})$ | |
| | NAG + OGM-G | [45] | $O(\frac{n}{\sqrt{\epsilon}})$ | $O(\frac{1}{\sqrt{\epsilon}} + d)$ memory, optimal in $\epsilon$ |
| | NAG + M-OGM-G | [Section 3.1] | $O(\frac{n}{\sqrt{\epsilon}})$ | $O(d)$ memory, optimal in $\epsilon$ |
| | Katyusha + L2S | [Appendix E] | $O(n \log \frac{1}{\epsilon} + \frac{\sqrt{n}}{\epsilon^{2/3}})$ | |
| | Acc-SVRG-G | [Section 4] | $O(n \log \frac{1}{\epsilon} + \frac{n^{2/3}}{\epsilon^{2/3}})$[1] | $O(n \log \frac{1}{\epsilon} + \sqrt{\frac{n}{\epsilon}})$ for function at the same time, simple and elegant |
| | Regularized Katyusha* | [2] | $O((n + \sqrt{\frac{n}{\epsilon}}) \log \frac{1}{\epsilon})$ | Requires the knowledge of $R_0$ |
| | R-Acc-SVRG-G* | [Section 5] | $O((n \log \frac{1}{\epsilon} + \sqrt{\frac{n}{\epsilon}}) \log \frac{1}{\epsilon})$ | Without the knowledge of $R_0$ |

* Indirect methods (using regularization).

Moreover, finding near-stationary points is a harder task than minimizing function value, because NAG has the optimal guarantee for $f(x) - f(x^\star)$ but is only suboptimal for minimizing $\|\nabla f(x)\|$.

In this work, we consider the problem $\min_{x \in \mathbb{R}^d} f(x) = \frac{1}{n} \sum_{i=1}^{n} f_i(x)$, where each $f_i$ is $L$-smooth and convex. We focus on finding an $\epsilon$-stationary point of this objective, i.e., a point with $\|\nabla f(x)\| \leq \epsilon$. We use $\mathcal{X}^\star$ to denote the set of optimal solutions, which is assumed to be nonempty. There are two different assumptions on the initial point $x_0$, namely, the Initial bounded-Function Condition (**IFC**): $f(x_0) - f(x^\star) \leq \Delta_0$, and the Initial bounded-Distance Condition (**IDC**): $\|x_0 - x^\star\| \leq R_0$ for some $x^\star \in \mathcal{X}^\star$. This subtlety results in drastically different best achievable rates as studied in [7, 22]. Below we categorize existing algorithmic techniques into three classes (relating to Table 1).

(i) *"IDC + IFC"*. Nesterov [42] showed that we can combine the guarantees of a method minimizing function value under IDC and a method finding near-stationary points under IFC to produce a faster one for minimizing gradient norm under IDC. For example, NAG produces $f(x_{K_1}) - f(x^\star) = O(\frac{L R_0^2}{K_1^2})$ [40] and GD produces $\|\nabla f(x_{K_2})\|^2 = O\big(\frac{L(f(x_0) - f(x^\star))}{K_2}\big)$ [33] under IFC. Letting $x_0 = x_{K_1}$ and $K = K_1 + K_2$, by balancing the ratio of $K_1$ and $K_2$, we obtain the guarantee $\|\nabla f(x_K)\|^2 = O(\frac{L^2 R_0^2}{K^3})$ for "NAG + GD". We point out that we can use this technique to combine the guarantees of Katyusha [1] and SARAH[2] [46]; see Appendix E.

(ii) *Regularization*. Nesterov [42] used NAG (strongly convex variant) to solve the regularized objective, and showed that it achieves near-optimal complexity (optimal up to logarithmic factors). Inspired by this technique, Allen-Zhu [2] proposed recursive regularization for stochastic approximation algorithms, which also achieves near-optimal complexities [22].

---

[1]Table 1 shows that Katyusha+L2S has a slightly better dependence on $n$ than Acc-SVRG-G. It is due to the adoption of $n$-dependent step size in L2S. As studied in [37], despite having a better complexity, $n$-dependent step size boosts numerical performance only when $n$ is *extremely large*. If the practically fast $n$-independent step size is used for L2S, Katyusha+L2S and Acc-SVRG-G have the same complexity. See also Appendix A.

[2]We adopt the loopless variant of SARAH in [37], which has a refined analysis for general convex objectives.

(iii) *Direct methods.* Due to the lack of insight, existing direct methods are mostly derived or analyzed with the help of computer-aided tools [31, 32, 54, 33]. The computer-aided approach was pioneered by Drori and Teboulle [19], who introduced the performance estimation problem (PEP). The only known optimal method OGM-G [33] was designed based on the PEP approach.

Observe that since $f(x) - f(x^\star) \leq \|\nabla f(x)\| \|x - x^\star\|$, the lower bound for finding near-stationary points must be of the same order as for minimizing function value [44]. Thus, under IDC, the lower bound is $\Omega(n + \sqrt{\frac{n}{\epsilon}})$ due to [58]. Under IFC, we can establish an $\Omega(n + \frac{\sqrt{n}}{\epsilon})$ lower bound using the techniques in [7, 58]. The main contributions of this work are three new algorithmic schemes that improve the practicalities of existing methods as summarized below (highlighted in Table 1).

- (Section 3) We propose a memory-saving variant of OGM-G for the deterministic case ($n = 1$), which does not require a pre-computed and stored parameter sequence. The derivation of the new variant is inspired by the numerical solution to a PEP problem.
- (Section 4) We propose a new accelerated SVRG [29, 59] variant that can *simultaneously* achieve fast convergence rates for minimizing both the gradient norm and function value, that is, $O(n \log \frac{1}{\epsilon} + \frac{n^{2/3}}{\epsilon^{2/3}})$ complexity for gradient norm and $O(n \log \frac{1}{\epsilon} + \sqrt{\frac{n}{\epsilon}})$ complexity for function value. Note that other stochastic approaches in Table 1 do not have this property.
- (Section 5) We propose an adaptively regularized accelerated SVRG variant, which does not require the knowledge of $R_0$ or $\Delta_0$ and achieves a near-optimal complexity under IDC or IFC.

We put in extra efforts to make the proposed schemes as simple and elegant as possible. We believe that the simplicity makes the extensions of the new schemes easier.

## 2   Preliminaries

Throughout this paper, we use $\langle \cdot, \cdot \rangle$ and $\|\cdot\|$ to denote the inner product and the Euclidean norm, respectively. We let $[n]$ denote the set $\{1, 2, \ldots, n\}$, $\mathbb{E}$ denote the total expectation and $\mathbb{E}_{i_k}$ denote the expectation with respect to a random sample $i_k$. We say that a function $f : \mathbb{R}^d \to \mathbb{R}$ is *L-smooth* if it has $L$-Lipschitz continuous gradients, i.e.,

$$\forall x, y \in \mathbb{R}^d, \|\nabla f(x) - \nabla f(y)\| \leq L \|x - y\|.$$

A continuously differentiable $f$ is called $\mu$-*strongly convex* if

$$\forall x, y \in \mathbb{R}^d, f(x) - f(y) - \langle \nabla f(y), x - y \rangle \geq \frac{\mu}{2} \|x - y\|^2.$$

Other equivalent definitions of these two assumptions can be found in the textbook [44]. The following is an important consequence of a function $f$ being $L$-smooth and convex:

$$\forall x, y \in \mathbb{R}^d, f(x) - f(y) - \langle \nabla f(y), x - y \rangle \geq \frac{1}{2L} \|\nabla f(x) - \nabla f(y)\|^2. \tag{1}$$

We call (1) the *interpolation condition* at $(x, y)$ following [56]. If $f$ is both $L$-smooth and $\mu$-strongly convex, we can define a "shifted" function $h(x) = f(x) - f(x^\star) - \frac{\mu}{2} \|x - x^\star\|^2$ following [63]. It can be easily verified that $h$ is $(L - \mu)$-smooth and convex, and thus from (1),

$$\forall x, y \in \mathbb{R}^d, h(x) - h(y) - \langle \nabla h(y), x - y \rangle \geq \frac{1}{2(L - \mu)} \|\nabla h(x) - \nabla h(y)\|^2, \tag{2}$$

which is equivalent to the *strongly convex interpolation condition* discovered in [56].

Oracle complexity (or simply complexity) refers to the required number of stochastic gradient $\nabla f_i$ computations to find an $\epsilon$-accurate solution.

## 3   OGM-G: "Momentum" Reformulation and a Memory-Saving Variant

In this section, we focus on the IFC case, i.e., $f(x_0) - f(x^\star) \leq \Delta_0$. We use $N$ to denote the total iteration number to prevent confusion (in other sections, we use $K$). Proofs in this section are given in

---

**Algorithm 1** OGM-G: "Momentum" reformulation

---

**Input:** initial guess $x_0 \in \mathbb{R}^d$, total iteration number $N$.
**Initialize:** vector $v_0 = \mathbf{0}$, scalars $\theta_N = 1$ and $\theta_k^2 - \theta_k = \theta_{k+1}^2$, for $k = 0 \ldots N - 1$.
  1: **for** $k = 0, \ldots, N - 1$ **do**
  2:    $v_{k+1} = v_k + \frac{1}{L\theta_k \theta_{k+1}^2} \nabla f(x_k)$.
  3:    $x_{k+1} = x_k - \frac{1}{L} \nabla f(x_k) - (2\theta_{k+1}^3 - \theta_{k+1}^2) v_{k+1}$.
  4: **end for**
**Output:** $x_N$.

---

Appendix B. Recall that OGM-G has the following updates [33]. Let $y_0 = x_0$. For $k = 0, \ldots, N-1$,

$$
\begin{aligned}
y_{k+1} &= x_k - \frac{1}{L}\nabla f(x_k), \\
x_{k+1} &= y_{k+1} + \frac{(\theta_k - 1)(2\theta_{k+1} - 1)}{\theta_k(2\theta_k - 1)}(y_{k+1} - y_k) + \frac{2\theta_{k+1} - 1}{2\theta_k - 1}(y_{k+1} - x_k),
\end{aligned}
\tag{3}
$$

where $\{\theta_k\}$ is recursively defined: $\theta_N = 1$ and $\begin{cases} \theta_k^2 - \theta_k = \theta_{k+1}^2 & k = 1 \ldots N - 1, \\ \theta_0^2 - \theta_0 = 2\theta_1^2 & \text{otherwise.} \end{cases}$

OGM-G was discovered from the numerical solution to an SDP problem and its analysis is to show that the step coefficients in (3) specify a feasible solution to the SDP problem. While this analysis is natural for the PEP approach, it is hard to understand how each coefficient affects the rate, especially if one wants to generalize the scheme. Here we provide a simple algebraic analysis for OGM-G.

We start with a reformulation[3] of OGM-G in Algorithm 1, which aims to simplify the proof. We adopt a consistent $\{\theta_k\}$: $\theta_N = 1$ and $\theta_k^2 - \theta_k = \theta_{k+1}^2, k = 0 \ldots N - 1$, which only costs a constant factor.[4] Interestingly, the reformulated scheme resembles the heavy-ball momentum method [49]. However, it can be shown that Algorithm 1 is not covered by the heavy-ball momentum scheme. Defining $\theta_{N+1}^2 = \theta_N^2 - \theta_N = 0$, we provide the one-iteration analysis in the following proposition:

**Proposition 3.1.** *In Algorithm 1, the following holds at any iteration* $k \in \{0, \ldots, N - 1\}$ :

$$
A_k + B_{k+1} + C_{k+1} + E_{k+1} \leq A_{k+1} + B_k + C_k + E_k - \theta_{k+1}\langle \nabla f(x_{k+1}), v_{k+1}\rangle
$$
$$
+ \sum_{i=k+1}^{N} \frac{\theta_i}{L\theta_k \theta_{k+1}^2}\langle \nabla f(x_k), \nabla f(x_i)\rangle,
\tag{4}
$$

*with* $A_k \triangleq \frac{1}{\theta_k^2}(f(x_N) - f(x^\star) - \frac{1}{2L}\|\nabla f(x_N)\|^2)$, $B_k \triangleq \frac{1}{\theta_k^2}(f(x_k) - f(x^\star))$, $C_k \triangleq \frac{1}{2L\theta_k^2}\|\nabla f(x_k)\|^2$,
$E_k \triangleq \frac{\theta_{k+1}^2}{\theta_k}\langle \nabla f(x_k), v_k\rangle$.

**Remark 3.1.1.** *A recent work [15] also conducted an algebraic analysis of OGM-G under a potential function framework. Their potential function decrease can be directly obtained from Proposition 3.1 by summing up (4). By contrast, our "momentum" vector $\{v_k\}$ naturally merges into the analysis, which significantly simplifies the analysis. Moreover, it provides a better interpretation on how OGM-G utilizes the past gradients to achieve acceleration.*

From (4), we see that only the last two terms do not telescope. Note that the "momentum" vector is a weighted sum of the past gradients, i.e., $v_{k+1} = \sum_{i=0}^{k} \frac{1}{L\theta_i \theta_{i+1}^2}\nabla f(x_i)$. If we sum the terms up from $k = 0, \ldots, N - 1$, it can be verified that they exactly sum up to 0. The presence of these special terms prevents OGM-G to have a usual potential function (e.g., those in [6]). Then, by telescoping the remaining terms, we obtain the final convergence guarantee.

**Theorem 3.1.** *The output of Algorithm 1 satisfies* $\|\nabla f(x_N)\|^2 \leq \frac{8L\Delta_0}{(N+2)^2}$.

We observe two drawbacks of OGM-G (same as the algorithm description in [15]): (1) it requires storing a pre-computed parameter sequence, which costs $O(\frac{1}{\epsilon})$ floats; (2) except for the last iterate,

---

[3]It can be verified that this scheme is equivalent to the original one (3) through $v_k = \frac{1}{(2\theta_k - 1)\theta_k^2}(y_k - x_k)$.

[4]The original guarantee of OGM-G can be recovered if we set $\theta_0^2 - \theta_0 = 2\theta_1^2$.

---

**Algorithm 2** M-OGM-G: Memory-saving OGM-G

---

**Input:** initial guess $x_0 \in \mathbb{R}^d$, total iteration number $N$.
**Initialize:** vector $v_0 = \mathbf{0}$.
 1: **for** $k = 0, \ldots, N - 1$ **do**
 2:      $v_{k+1} = v_k + \frac{12}{L(N-k+1)(N-k+2)(N-k+3)} \nabla f(x_k).$
 3:      $x_{k+1} = x_k - \frac{1}{L} \nabla f(x_k) - \frac{(N-k)(N-k+1)(N-k+2)}{6} v_{k+1}.$
 4: **end for**
**Output:** $x_N$ or $\arg\min_{x \in \{x_0, \ldots, x_N\}} \|\nabla f(x)\|.$

---

123 all other iterates are not known to have guarantees. We resolve these issues by proposing another
124 parameterization of Algorithm 1 in the next subsection.

## 3.1 Memory-Saving OGM-G

126 A straightforward idea to resolve the aforementioned issues is to generalize Algorithm 1. However,
127 we find it rather difficult since the parameters in the analysis are rather strict (despite that the proof is
128 already simple). We choose to rely on computer-aided techniques [19]. The derivation of this variant
129 (Algorithm 2) is based on the following numerical experiment.

130 **Numerical experiment.** OGM-G was discovered when considering the relaxed PEP problem [33]:

$$\max_{\substack{\nabla f(x_0), \ldots, \nabla f(x_N) \in \mathbb{R}^d \\ f(x_0), \ldots, f(x_N), f(x^\star) \in \mathbb{R}}} \|\nabla f(x_N)\|^2$$

$$\text{subject to} \begin{cases} \text{interpolation condition (1) at } (x_k, x_{k+1}), & k = 0, \ldots, N-1, \\ \text{interpolation condition (1) at } (x_N, x_k), & k = 0, \ldots, N-1, \\ \text{interpolation condition (1) at } (x_N, x^\star), & f(x_0) - f(x^\star) \leq \Delta_0, \end{cases} \tag{P}$$

131 where the sequence $\{x_k\}$ is defined as $x_{k+1} = x_k - \frac{1}{L} \sum_{i=0}^k h_{k+1,i} \nabla f(x_i), k = 0, \ldots, N-1$ for
132 some step coefficients $h \in \mathbb{R}^{N(N+1)/2}$. Given $N$, the step coefficients of OGM-G correspond to
133 a numerical solution to the problem: $\arg\min_h \{\text{Lagrangian dual of (P)}\}$, which is denoted as (HD).
134 Conceptually, solving problem (HD) would give us the fastest possible step coefficients under the
135 constraints.[5] We expect there to be some constant-time slower schemes, which are neglected when
136 solving (HD). To identify such schemes, we relax a set of interpolation conditions in problem (P):

$$f(x_N) - f(x_k) - \langle \nabla f(x_k), x_N - x_k \rangle \geq \frac{1}{2L} \|\nabla f(x_N) - \nabla f(x_k)\|^2 - \rho \|\nabla f(x_k)\|^2,$$

137 for $k = 0, \ldots, N - 1$ and some $\rho > 0$. After this relaxation, solving (HD) will no longer give us the
138 step coefficients of OGM-G. By trying different $\rho$ and checking the dependence on $N$, we discover
139 Algorithm 2 when $\rho = \frac{1}{2L}$. Similar to our analysis of OGM-G, we provide a simple algebraic analysis
140 for the new variant in the following theorem.

141 **Theorem 3.2.** *Define* $\delta_{k+1} \triangleq \frac{12}{(N-k+1)(N-k+2)(N-k+3)}, k = 0, \ldots, N$. *In Algorithm 2, it holds that*
142

$$\sum_{k=0}^N \frac{\delta_{k+1}}{2} \|\nabla f(x_k)\|^2 \leq \frac{12L\Delta_0}{(N+2)(N+3)}. \tag{5}$$

143 **Remark 3.2.1.** *Algorithm 2 converges optimally on the last iterate (note that $\delta_{N+1} = 2$) and the*
144 *minimum gradient since*

$$\min_{k \in \{0, \ldots, N\}} \|\nabla f(x_k)\|^2 \leq \frac{1}{\sum_{k=0}^N \frac{\delta_{k+1}}{2}} \sum_{k=0}^N \frac{\delta_{k+1}}{2} \|\nabla f(x_k)\|^2 \leq \frac{8L\Delta_0}{(N+2)(N+3)-2}.$$

145 Clearly, the parameters of this variant can be computed on the fly and from (5), each iterate has a
146 guarantee (although the guarantee degenerates quickly as $k \to 0$ since $1/\delta_{k+1} = \Omega((N-k)^3)$).
147 Moreover, we can extend the benefits into the IDC case using the ideas in [42] as summarized below.

---

[5]However, since problem (HD) is non-convex, we can only obtain approximate solutions.

---

---
**Algorithm 3** Acc-SVRG-G: Accelerated SVRG for Gradient minimization
---
**Input:** parameters $\{\tau_k\}$, $\{p_k\}$, initial guess $x_0 \in \mathbb{R}^d$, total iteration number $K$.

**Initialize:** vectors $z_0 = \tilde{x}_0 = x_0$ and scalars $\alpha_k = \frac{L\tau_k}{1-\tau_k}, \forall k$ and $\tilde{\tau} = \sum_{k=0}^{K-1} \tau_k^{-2}$.

1: **for** $k = 0, \ldots, K-1$ **do**

2: $\quad y_k = \tau_k z_k + (1 - \tau_k)\left(\tilde{x}_k - \frac{1}{L}\nabla f(\tilde{x}_k)\right)$.

3: $\quad z_{k+1} = \arg\min_x \left\{ \langle \mathcal{G}_k, x \rangle + (\alpha_k/2)\|x - z_k\|^2 \right\}$.

4: $\quad // \mathcal{G}_k \triangleq \nabla f_{i_k}(y_k) - \nabla f_{i_k}(\tilde{x}_k) + \nabla f(\tilde{x}_k)$, where $i_k$ is sampled uniformly in $[n]$.

5: $\quad \tilde{x}_{k+1} = \begin{cases} y_k & \text{with probability } p_k, \\ \tilde{x}_k & \text{with probability } 1 - p_k. \end{cases}$

6: **end for**

**Output (for gradient):** $x_{\text{out}}$ is sampled from $\left\{ \text{Prob}\{x_{\text{out}} = \tilde{x}_k\} = \frac{\tau_k^{-2}}{\tilde{\tau}} \, \middle| \, k \in \{0, \ldots, K-1\} \right\}$.

**Output (for function value):** $\tilde{x}_K$.

---

**Corollary 3.2.1** (IDC case)**.** *If we first run $N/2$ iterations of NAG and then continue with $N/2$ iterations of Algorithm 2, we obtain an output satisfying $\|\nabla f(x_N)\| = O(\frac{LR_0}{N^2})$.*

## 4 Accelerated SVRG: Fast Rates for Both Gradient Norm and Objective

In this section, we focus on the IDC case, i.e., $\|x_0 - x^\star\| \leq R_0$ for some $x^\star \in \mathcal{X}^\star$. From the development in the previous section, it is natural to ask whether we can use the PEP approach to motivate new stochastic schemes. However, due to the exponential growth of the number of possible states $(i_0, i_1, \ldots)$, we cannot directly adopt this approach. A feasible alternative is to first fix an algorithmic framework and a family of potential functions, and then use the potential-based PEP approach in [54]. However, this approach is much more restrictive. For example, it cannot identify special constructions like (4) in OGM-G. Fortunately, as we will see, we can get some inspiration from the recent development of deterministic methods. Proofs in this section are given in Appendix C.

Our proposed scheme is given in Algorithm 3. We adopt the elegant loopless design of SVRG in [34]. Note that the full gradient $\nabla f(\tilde{x}_k)$ is computed and stored only when $\tilde{x}_{k+1} = y_k$ at Step 5. We summarize our main technical novelty as follows.

**Main algorithmic novelty.** The design of stochastic accelerated methods is largely inspired by NAG. To make it clear, by setting $n = 1$, we see that Katyusha [1], MiG [61], SSNM [62], Varag [36], VRADA [52], ANITA [38], the acceleration framework in [16] and AC-SA [35, 24] all reduce to one of the following variants of NAG. We say that these methods are under the NAG framework.

$$\begin{cases} x_k = \tau_k z_k + (1 - \tau_k)y_k, \\ z_{k+1} = z_k - \alpha_k \nabla f(x_k), \\ y_{k+1} = \tau_k z_{k+1} + (1 - \tau_k)y_k. \end{cases} \qquad \begin{cases} x_k = \tau_k z_k + (1 - \tau_k)y_k, \\ z_{k+1} = z_k - \alpha_k \nabla f(x_k), \\ y_{k+1} = x_k - \eta_k \nabla f(x_k). \end{cases}$$

$$\text{Auslender and Teboulle [5]} \qquad\qquad \text{Linear Coupling [64]}$$

See [57, 12] for other variants of NAG. When $n = 1$, Algorithm 3 reduces to the following scheme:

$$\begin{cases} y_k = \tau_k z_k + (1 - \tau_k)\left(y_{k-1} - \frac{1}{L}\nabla f(y_{k-1})\right), \\ z_{k+1} = z_k - \frac{1}{\alpha_k}\nabla f(y_k). \end{cases}$$

$$\text{Optimized Gradient Method (OGM) [19, 30]}$$

Algorithm 3 reduces to the scheme of OGM when $n = 1$ (this point is clearer in the formulation of ITEM in [55]). OGM has a constant-time faster worst-case rate than NAG, which exactly matches the lower complexity bound established in [17]. In the following proposition, we show that the OGM framework helps us conduct a tight one-iteration analysis, which gives room for achieving our goal.

**Proposition 4.1.** *In Algorithm 3, the following holds at any iteration $k \geq 0$ and $\forall x^\star \in \mathcal{X}^\star$:*

$$
\left( \frac{1 - \tau_k}{\tau_k^2 p_k} \mathbb{E}\left[ f(\tilde{x}_{k+1}) - f(x^\star) \right] + \frac{L}{2} \mathbb{E}\left[ \|z_{k+1} - x^\star\|^2 \right] \right) + \frac{(1 - \tau_k)^2}{2L\tau_k^2} \mathbb{E}\left[ \|\nabla f(\tilde{x}_k)\|^2 \right]
$$
$$
\leq \left( \frac{(1 - \tau_k p_k)(1 - \tau_k)}{\tau_k^2 p_k} \mathbb{E}\left[ f(\tilde{x}_k) - f(x^\star) \right] + \frac{L}{2} \mathbb{E}\left[ \|z_k - x^\star\|^2 \right] \right). \tag{6}
$$

The terms inside the parentheses form the commonly used potential function of SVRG variants. The additional $\mathbb{E}[\|\nabla f(\tilde{x}_k)\|^2]$ term is created by adopting the OGM framework. In other words, we use the following potential function for Algorithm 3 ($a_k, b_k, c_k \geq 0$):

$$
T_k = a_k \mathbb{E}\left[ f(\tilde{x}_k) - f(x^\star) \right] + b_k \mathbb{E}\left[ \|z_k - x^\star\|^2 \right] + \sum_{i=0}^{k-1} c_i \mathbb{E}\left[ \|\nabla f(\tilde{x}_i)\|^2 \right].
$$

We first provide a simple parameter choice, which leads to a simple and clean analysis.

**Theorem 4.1** (Single-stage parameter choice)**.** *In Algorithm 3, if we choose $p_k \equiv \frac{1}{n}, \tau_k = \frac{3}{k/n+6}$, then the following holds at the outputs:*

$$
\mathbb{E}\left[ \|\nabla f(x_{\text{out}})\|^2 \right] = O\left( \frac{n^3 L\big(f(x_0) - f(x^\star)\big) + n^2 L^2 R_0^2}{K^3} \right),
$$
$$
\mathbb{E}\left[ f(\tilde{x}_K) \right] - f(x^\star) = O\left( \frac{n^2 \big(f(x_0) - f(x^\star)\big) + nLR_0^2}{K^2} \right). \tag{7}
$$

*In other words, to guarantee that $\mathbb{E}\left[ \|\nabla f(x_{\text{out}})\| \right] \leq \epsilon_g$ and $\mathbb{E}\left[ f(\tilde{x}_K) \right] - f(x^\star) \leq \epsilon_f$, the oracle complexities are $O\left( \frac{n(L(f(x_0) - f(x^\star)))^{1/3}}{\epsilon_g^{2/3}} + \frac{(nLR_0)^{2/3}}{\epsilon_g^{2/3}} \right)$ and $O\left( n\sqrt{\frac{f(x_0) - f(x^\star)}{\epsilon_f}} + \frac{\sqrt{nL}R_0}{\sqrt{\epsilon_f}} \right)$, respectively.*

From (7), we see that Algorithm 3 achieves fast $O(\frac{1}{K^{1.5}})$ and $O(\frac{1}{K^2})$ rates for minimizing the gradient norm and function value at the same time. However, despite being a simple choice, the oracle complexities are not better than the deterministic methods in Table 1. Below we provide a two-stage parameter choice, which is inspired by the idea of including a "warm-up phase" in [3, 36, 52, 38]. This theorem corresponds to the reported result in Table 1.

**Theorem 4.2** (Two-stage parameter choice)**.** *In Algorithm 3, let $p_k = \max\{\frac{6}{k+8}, \frac{1}{n}\}, \tau_k = \frac{3}{p_k(k+8)}$. The oracle complexities needed to guarantee $\mathbb{E}\left[ \|\nabla f(x_{\text{out}})\| \right] \leq \epsilon_g$ and $\mathbb{E}\left[ f(\tilde{x}_K) \right] - f(x^\star) \leq \epsilon_f$ are*

$$
O\left( n \min\left\{ \log \frac{LR_0}{\epsilon_g}, \log n \right\} + \frac{(nLR_0)^{2/3}}{\epsilon_g^{2/3}} \right) \text{ and } O\left( n \min\left\{ \log \frac{LR_0^2}{\epsilon_f}, \log n \right\} + \frac{\sqrt{nL}R_0}{\sqrt{\epsilon_f}} \right),
$$

*respectively.*

If $\epsilon$ is large or $n$ is very large, the recently proposed ANITA [38] achieves an $O(n)$ complexity, which matches the lower complexity bound $\Omega(n)$ in this case [58]. Since ANITA uses the NAG framework, we show that similar results can be derived under the OGM framework in the following theorem:

**Theorem 4.3** (Low accuracy parameter choice)**.** *In Algorithm 3, let iteration $N$ be the first time Step 5 updates $\tilde{x}_{k+1} = y_k$. If we choose $p_k \equiv \frac{1}{n}, \tau_k \equiv 1 - \frac{1}{\sqrt{n+1}}$ and terminate Algorithm 3 at iteration $N$, then the following holds at $\tilde{x}_{N+1}$:*

$$
\mathbb{E}\left[ \|\nabla f(\tilde{x}_{N+1})\|^2 \right] \leq \frac{8L^2 R_0^2}{5(\sqrt{n+1}+1)} \text{ and } \mathbb{E}\left[ f(\tilde{x}_{N+1}) \right] - f(x^\star) \leq \frac{LR_0^2}{\sqrt{n+1}+1}.
$$

*In particular, if the required accuracies are low (or $n$ is very large), i.e., $\epsilon_g^2 \geq \frac{8L^2 R_0^2}{5(\sqrt{n+1}+1)}$ and $\epsilon_f \geq \frac{LR_0^2}{\sqrt{n+1}+1}$, then Algorithm 3 only has an $O(n)$ oracle complexity.*

In the low accuracy region (specified above), the choice in Theorem 4.3 removes the $O(\log \frac{1}{\epsilon})$ factor in the complexity of Theorem 4.2. We include some numerical justifications of Algorithm 3 in Appendix A. We believe that the potential-based PEP approach in [54] can help us identify better parameter choices of Algorithm 3, which we leave for future work.

---

**Algorithm 4** R-Acc-SVRG-G

---

**Input:** accuracy $\epsilon > 0$, parameters $\delta_0 = L, \beta > 1$, initial guess $x_0 \in \mathbb{R}^d$.

1: **for** $t = 0, 1, 2, \ldots$ **do**
2:     Define $f^{\delta_t}(x) = (1/n) \sum_{i=1}^n f_i^{\delta_t}(x)$, where $f_i^{\delta_t}(x) = f_i(x) + (\delta_t/2) \|x - x_0\|^2$.
3:     Initialize vectors $z_0 = \tilde{x}_0 = x_0$ and set $\tau_x, \tau_z, \alpha, p, C_{\text{IDC}}, C_{\text{IFC}}$ according to Proposition 5.1.
4:     **for** $k = 0, 1, 2, \ldots$ **do**
5:         $y_k = \tau_x z_k + (1 - \tau_x) \tilde{x}_k + \tau_z \left( \delta_t(\tilde{x}_k - z_k) - \nabla f^{\delta_t}(\tilde{x}_k) \right)$.
6:         $z_{k+1} = \arg\min_x \left\{ \left\langle \mathcal{G}_k^{\delta_t}, x \right\rangle + (\alpha/2) \|x - z_k\|^2 + (\delta_t/2) \|x - y_k\|^2 \right\}$.
7:         $/\!/ \mathcal{G}_k^{\delta_t} \triangleq \nabla f_{i_k}^{\delta_t}(y_k) - \nabla f_{i_k}^{\delta_t}(\tilde{x}_k) + \nabla f^{\delta_t}(\tilde{x}_k)$, where $i_k$ is sampled uniformly in $[n]$.
8:         $\tilde{x}_{k+1} = \begin{cases} y_k & \text{with probability } p, \\ \tilde{x}_k & \text{with probability } 1 - p. \end{cases}$
9:         **if** [6]$\|\nabla f(\tilde{x}_k)\| \leq \epsilon$ **then** output $\tilde{x}_k$ and terminate the algorithm.
10:         **if** under IDC and $(1 + \frac{\delta_t}{\alpha})^k \geq \sqrt{C_{\text{IDC}}}/\delta_t$ **then** break the inner loop.
11:         **if** under IFC and $(1 + \frac{\delta_t}{\alpha})^k \geq \sqrt{C_{\text{IFC}}/2\delta_t}$ **then** break the inner loop.
12:     **end for**
13:     $\delta_{t+1} = \delta_t/\beta$.
14: **end for**

---

## 5   Near-Optimal Accelerated SVRG with Adaptive Regularization

Currently, there is no known stochastic method that directly achieves the optimal rate in $\epsilon$. To get near-optimal rates, the existing strategy is to use a carefully designed regularization technique [42, 2] with a method that solves strongly convex problems; see, e.g., [42, 2, 22, 11]. However, the regularization parameter requires the knowledge of $R_0$ or $\Delta_0$, which significantly limits its practicality.

Inspired by the recently proposed adaptive regularization technique [27], we develop a near-optimal accelerated SVRG variant (Algorithm 4) that does not require the knowledge of $R_0$ or $\Delta_0$. Note that this technique was originally proposed for NAG under the IDC assumption. Our development extends this technique to the stochastic setting, which brings an $O(\sqrt{n})$ rate improvement. Moreover, we consider both IFC and IDC cases. Proofs in this section are provided in Appendix D.

**Detailed design.** Algorithm 4 has a "guess-and-check" framework. In the outer loop, we first define the regularized objective $f^{\delta_t}$ using the current estimate of regularization parameter $\delta_t$, and then we initialize an accelerated SVRG method (the inner loop) to solve the $\delta_t$-strongly convex $f^{\delta_t}$. If the inner loop breaks at Step 10 or 11, indicating the poor quality of the current estimate $\delta_t$, $\delta_t$ will be divided by a fixed $\beta$. Thus, conceptually, we can adopt any method that solves strongly convex finite-sums at the optimal rate as the inner loop. However, since the constructions of Step 10 or 11 require some algorithm-dependent constants, we have to fix one method as the inner loop.

The inner loop we adopted is a loopless variant of BS-SVRG [63]. This is because (i) BS-SVRG is the fastest known accelerated SVRG variant (for ill-conditioned problems) and (ii) it has a simple scheme, especially after using the loopless construction [34]. However, its original guarantee is built upon $\{z_k\}$. Clearly, we cannot implement the stopping criterion (Step 9) on $\|\nabla f(z_k)\|$. Interestingly, we discover that its sequence $\{\tilde{x}_k\}$ works perfectly in our regularization framework, even if we can neither establish convergence on $f(\tilde{x}_k) - f(x^\star)$ nor on $\|\tilde{x}_k - x^\star\|^2$.[7] Moreover, we find that the loopless construction significantly simplifies the parameter constraints of BS-SVRG, which originally involves $\Theta(n)$th-order inequality. We provide the detailed parameter choice as follows:

**Proposition 5.1** (Parameter choice). *In Algorithm 4, we set* $\tau_x = \frac{\alpha + \delta_t}{\alpha + L + \delta_t}, \tau_z = \frac{\tau_x}{\delta_t} - \frac{\alpha(1 - \tau_x)}{\delta_t L}$ *and* $p = \frac{1}{n}$. *We set* $\alpha$ *as the (unique) positive root of the cubic equation* $\left( 1 - \frac{p(\alpha + \delta_t)}{\alpha + L + \delta_t} \right) \left( 1 + \frac{\delta_t}{\alpha} \right)^2 = 1$ *and specify* $C_{\text{IDC}} = L^2 + \frac{L\alpha^2 p}{L + (1-p)(\alpha + \delta_t)}, C_{\text{IFC}} = 2L + \frac{2L\alpha^2 p}{(L + (1-p)(\alpha + \delta_t))\delta_t}$. *Under these choices, we have* $\frac{\alpha}{\delta_t} = O\left(n + \sqrt{n(L/\delta_t + 1)}\right), C_{\text{IDC}} = O\left((L + \delta_t)^2\right)$, *and* $C_{\text{IFC}} = O(L)$.

---

[6]Note that we maintain the full gradient $\nabla f^{\delta_t}(\tilde{x}_k)$ and $\nabla f(\tilde{x}_k) = \nabla f^{\delta_t}(\tilde{x}_k) - \delta_t(\tilde{x}_k - x_0)$.

[7]It is due to the special potential function of BS-SVRG (see (27)), which does not contain these two terms.

Under the choices of $\tau_x$ and $\tau_z$, the $\alpha$ above is the optimal choice in our analysis. Then, we can characterize the progress of the inner loop in the following proposition:

**Proposition 5.2** (The inner loop of Algorithm 4). *Using the parameters specified in Proposition 5.1, after running the inner loop (Step 4-12) of Algorithm 4 for k iterations, we can conclude that*

*(i) under IDC, i.e., $\|x_0 - x^\star\| \leq R_0$ for some $x^\star \in \mathcal{X}^\star$,*

$$\mathbb{E}\left[\|\nabla f(\tilde{x}_k)\|\right] \leq \left(\delta_t + \left(1 + \frac{\delta_t}{\alpha}\right)^{-k}\sqrt{C_{\text{IDC}}}\right)R_0,$$

*(ii) under IFC, i.e., $f(x_0) - f(x^\star) \leq \Delta_0$,*

$$\mathbb{E}\left[\|\nabla f(\tilde{x}_k)\|\right] \leq \left(\sqrt{2\delta_t} + \left(1 + \frac{\delta_t}{\alpha}\right)^{-k}\sqrt{C_{\text{IFC}}}\right)\sqrt{\Delta_0}.$$

The above results motivate the design of Step 10 and 11. For example, in the IDC case, when the inner loop breaks at Step 10, using *(i)* above, we obtain $\mathbb{E}\left[\|\nabla f(\tilde{x}_k)\|\right] \leq 2\delta_t R_0$. Then, by discussing the relative size of $\delta_t$ and a certain constant, we can estimate the complexity of Algorithm 4. The same methodology is used for the IFC case.

**Theorem 5.1** (IDC case). *Denote $\delta_{\text{IDC}}^\star = \frac{\epsilon q}{2R_0}$ for some $q \in (0, 1)$ and let the outer iteration $t = \ell$ be the first time[8] $\delta_\ell \leq \delta_{\text{IDC}}^\star$. The following assertions hold:*

*(i) At outer iteration $\ell$, Algorithm 4 terminates with probability at least $1 - q$.[9]*

*(ii) The total expected oracle complexity of the $\ell + 1$ outer loops is*

$$O\left(\left(n\log\frac{LR_0}{\epsilon q} + \sqrt{\frac{nLR_0}{\epsilon q}}\right)\log\frac{LR_0}{\epsilon q}\right).$$

**Theorem 5.2** (IFC case). *Denote $\delta_{\text{IFC}}^\star = \frac{\epsilon^2 q^2}{8\Delta_0}$ for some $q \in (0, 1)$ and let the outer iteration $t = \ell$ be the first time $\delta_\ell \leq \delta_{\text{IFC}}^\star$. The following assertions hold:*

*(i) At outer iteration $\ell$, Algorithm 4 terminates with probability at least $1 - q$.*

*(ii) The total expected oracle complexity of the $\ell + 1$ outer loops is*

$$O\left(\left(n\log\frac{\sqrt{L\Delta_0}}{\epsilon q} + \frac{\sqrt{nL\Delta_0}}{\epsilon q}\right)\log\frac{\sqrt{L\Delta_0}}{\epsilon q}\right).$$

Compared with regularized Katyusha in Table 1, the adaptive regularization approach drops the need to estimate $R_0$ or $\Delta_0$ at the cost of a mere $\log\frac{1}{\epsilon}$ factor in the non-dominant term (if $\epsilon$ is small).

# 6 Discussion

In this work, we proposed several simple and practical schemes that complement existing works (Table 1). Admittedly, the new schemes are currently only limited to the unconstrained Euclidean setting, because our techniques heavily rely on the interpolation conditions (1) and (2). On the other hand, methods such as OGM [30], TM [51] and ITEM [55, 10], which also rely on these conditions, are still not known to have their proximal variants. We list a few future directions as follows.

(1) It is not clear how to naturally connect the parameters of M-OGM-G (Algorithm 2) to OGM-G (Algorithm 1). The parameters of both algorithms seem to be quite restrictive and hardly generalizable due to the special construction in (4). Does there exist an optimal method for minimizing the gradient norm that has a proper potential function (at each iteration)?

(2) Is this new "momentum" in OGM-G beneficial for training neural nets? Other classic momentum schemes such as NAG [40] or heavy-ball momentum method [49] are extremely effective for this task [53], and they were also originally proposed for convex objectives.

(3) Can we directly accelerate SARAH (L2S)? By extending OGM-G? It seems that existing stochastic acceleration techniques fail to accelerate SARAH (or result in poor dependence on $n$ as in [16]).

---

[8]We assume that $\epsilon$ is small such that $\max\{\delta_{\text{IDC}}^\star, \delta_{\text{IFC}}^\star\} \leq \delta_0 = L$ for simplicity. In this case, $\ell > 0$.

[9]If Algorithm 4 does not terminate at outer iteration $\ell$, it terminates at the next outer iteration with probability at least $1 - q/\beta$. That is, it terminates with higher and higher probability. The same goes for the IFC case.

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
