# OpenReview forum: "Practical Schemes for Finding Near-Stationary Points of Convex Finite-Sums"
_NeurIPS.cc/2021/Conference — NeurIPS 2021 Submitted_

### Official Review · Reviewer_3GpM · 2021-07-12

**Rating:** 9
**Confidence:** 2

**Summary:**

This paper proposes several optimization algorithms: 1. a memory-saving variant of OGM-G (by the way, could the authors provide the full name of OGM-G?); 2, a new accelerated SVRG (full name?); 3, Near-Optimal Accelerated SVRG with Adaptive Regularization. Analysis of convergence rate is provided.

**Limitations And Societal Impact:**

Limitations: not seen

Societal Impact: solid improvement of the optimization algorithms discussed, and therefore a better service to different application scenarios.

**Main Review:**

For the vanilla OGM-G, a reformulation is provided, and the convergence of the norm of gradient with the speed of O(1/N) is derived.

For the Memory-Saving OGM-G, the convergence rate O(1/N) is obtained for the minimum norm of gradient across the iterations 1, ..., N.

For accelerated SVRG, in terms of the norm of gradient, convergence rate O(1/N^1.5) is derived; in terms of function value, the convergence rate of O(1/N) is derived. (is K in this section the count N in previous sections? Would it be better to unify the notations?)

For Near-Optimal Accelerated SVRG with Adaptive Regularization, analysis is done around two initial value conditions, the first is Initial bounded-Function Condition (IFC), i.e., the function value at the initial guess is not too far away from the global minimum; the second condition is Initial bounded-Distance Condition (IDC), i.e, the initial guess is not too far away from one of the global optimizers.

**Time Spent Reviewing:**

4 hours

---

> ### Author Response · Authors · 2021-08-08
> **Reply To Reviewer 3GpM**
>
> Thank you for taking the time to carefully read our paper and all your constructive comments. We would like to address your comments as follows.
>
> 1. "The full names of OGM-G and SVRG".
>
>     The full name of OGM-G is Optimized Gradient Method for Gradient, and the full name of SVRG is Stochastic Variance Reduced Gradient method. We will add them to the paper.
>
> 2. "Is K in this section the count N in previous sections? Would it be better to unify the notations?".
>
>     Thanks for pointing this out. Originally we thought that the notation $\frac{(K-k)(K-k+1)(K-k+2)}{6}$ looks confusing in Algorithm 2, and thus we use $N$ instead of $K$ in Section 3. Now we realize that it is better to have different notations in Section 3 and Section 4. It is because in Section 3, $N$ denotes the number of full-gradient iterations each costs $O(n)$ $\nabla f_i(\cdot)$ computations, while in Section 4, $K$ denotes the number of stochastic iterations each costs $O(1)$ $\nabla f_i(\cdot)$ computations. We will clarify this in the paper.
>
> 3. About IFC and IDC conditions.
>
>     These two conditions are required by almost all the first-order methods (sometimes implicitly). As proved in Appendix A.2 in [1], if without IDC, any first-order method cannot efficiently minimize the function value in the worst case. Thus, IDC is a fundamental requirement of deriving convergence guarantees for minimizing the function value.
>
> ## Reference
> [1] Y. Carmon, J. C. Duchi, O. Hinder, and A. Sidford (2021). Lower bounds for finding stationary points ii: first-order methods. Mathematical Programming, 185(1-2).

---

### Official Review · Reviewer_ewr9 · 2021-07-16

**Rating:** 7
**Confidence:** 4

**Summary:**

This paper studies efficient methods for finding near-stationary points ($||\nabla f(x)|| \le \epsilon$) of a convex (finite-sum) function. The authors nicely introduce main reasons for the need of such research, and their three main contributions are as follows.

First, this paper provides variants of OGM-G that is an optimal method for finding a near-stationary point for a smooth convex function under the initial bounded-function condition (IFC). In specific, Proposition 3.1 complements the complexity of OGM-G (with and without a change of coefficient at the last iteration). In addition, a memory-saving variant of OGM-G was studied, which does not require computing coefficients in advance.

Second, based on OGM and the loopless SVRG, the authors propose Acc-SVRG-G. With the two-stage parameter choice under the initial bounded-distance condition (IDC), it has fast rates for minimizing both the gradient norm and the function value for the first time.

Third, with adaptive regularization, the Acc-SVRG-G achieves the best existing rates (by the regularized Katyusha) up to a logarithmic factor, in terms of minimizing the gradient norm, under both IDC and IFC. The proposed R-Acc-SVRG-G has advantage over the regularized Katyusha, since it does not require the knowledge of $R_0$ and $\Delta_0$.

**Limitations And Societal Impact:**

- In Section 4, since previous section studied OGM-G for finding near-stationary point, I was expecting the Acc-SVRG-G to be built upon (or related to) OGM-G (other than NAG and OGM). Also, while Section 3 mainly assumes IFC (then IDC), Section 4 mainly assumes IDC. Although the resulting analysis is interesting, I found Sections 3 and 4 to be somewhat disconnected.

- The parameters for R-Acc-SVRG-G is relatively complicated, but this seems inevitable.

- A numerical experiment in comparison to L2S, SAG, SVRG in Appendix A looks promising, but it is not sufficient in the eyes of practitioners. The authors did not compare with other algorithms that require fixing the accuracy $\epsilon$, but still one can compare methods in terms of the number of computational complexity to achieve the accuracy. (Note that I am not asking for further experiments, since the paper is theory-oriented.)

**Main Review:**

As stated in the beginning of the abstract, finding near-stationary points of convex optimization has not received attention, although it is fundamentally important. I think this paper nicely fills in the gap. In specific, developing variants of OGM-G is a nice complement to OGM-G, especially Proposition 3.1 and the relaxation of (P). A more interesting contribution of this paper is the development of both Acc-SVRG-G and R-Acc-SVRG-G. The latter achieves the best existing rates up to a logarithmic factor, while not requiring the knowledge of the (usually unavailable) initial constant, which is new and important.

I only have couple of minor comments.
- line 37: NAG having a suboptimal rate for minimizing the gradient does not seem to be a sufficient reason for saying that finding near-stationary point is a "harder" task.
- line 46: NAG + OGM-G is missing in the "IDC + IFC" case, unlike the Katyusha + L2S. Its variant is later mentioned in Corollary 3.2.1, so it would be nice to mention NAG + OGM-G at some point.
- line 73: I am curious here whether the function value complexity of Acc-SVRG-G is comparable to the best existing complexity.
- line 118, 254, 258: Note that a very recent paper below (appeared after the submission deadline) found a potential function for OGM-G and its proximal variant.
J. Lee, C. Park, E. K. Ryu, ``"A geometric structure of acceleration and its role in making gradients small fast,'' arXiv:2106.10439.
- line 123: OGM-G only has a convergence guarantee on the last iterate, and the authors claim that this can be resolved by M-OGM-G. However, Theorem 3.2 with a bound on the weighted sum of all iterates does not fully resolve the issue of OGM-G as I hoped from the line 123.
- line 135: In footnote 5, shouldn't it be local solutions, rather than approximate solutions?
- line 143: What do you mean by "Algorithm 2 converges optimally on the last iterate"?
- line 149: It is minor, but it would be nice to have a square in the gradient norm, so that it is consistent with the IFC case.
- line 177: $f(x_0) - f(x^*)$ -> $\Delta_0$?
- line 183: Is there any intuition behind the "warm-up phase", besides the two-stage proof in Appendix? This isn't used for R-Acc-SVRG-G. Would it be useful?
- line 191: Is the analysis for the low accuracy regime meaningful? (I might just being not familiar with such analysis.) Does Acc-SVRG-G with different parameter choice or any other existing method (like Katyusha) have a larger complexity for such regime?
- line 208: Which rate should I compare here?
- line 213: I kind of understand that such termination conditions indicate the poor quality of the current estimate $\delta_t$. Could you explain it in more details?

**Time Spent Reviewing:**

7

---

> ### Author Response · Authors · 2021-08-08
> **Reply To Reviewer ewr9**
>
> Thank you for taking the time to carefully read our paper and your detailed comments. We would like to address your comments as follows.
>
> 1. "I am curious here whether the function value complexity of Acc-SVRG-G is comparable to the best existing complexity".
>
>    The function value complexities of existing methods are (we use the citation numbers in the submission):
>    - Katyusha with AdaptReg [1]: $O(n\log{\frac{1}{\epsilon}} + \sqrt{\frac{n}{\epsilon}})$.
>    - Varag [36]: $O(n\min\lbrace\log{\frac{1}{\epsilon}}, \log{n}\rbrace + \sqrt{\frac{n}{\epsilon}})$. Acc-SVRG-G achieves the same rate in Theorem 4.2.
>    - VRADA [52]: $O(n\log\log{n} + \sqrt{\frac{n}{\epsilon}})$
>    - ANITA [38]: $O(n\min\lbrace 1+\log{\frac{1}{\epsilon\sqrt{n}}}, \log{\sqrt{n}}\rbrace + \sqrt{\frac{n}{\epsilon}})$, it's kind of confusing writing in this way. The main improvement of ANITA is the $O(n)$ complexity for the case $\epsilon \geq O(\frac{1}{\sqrt{n}})$ (identical to Acc-SVRG-G in Theorem 4.3); for the other cases, the rate is similar to Varag.
>
>    Thus, the best existing complexity is achieved by ANITA except for the case $O(\frac{1}{\sqrt{n}})\geq \epsilon \geq O(\frac{1}{n})$ (roughly speaking), in which VRADA is faster. The complexity of Acc-SVRG-G is similar to ANITA.
>
>    Clearly, Katyusha is already optimal up to log factors. This line of work aims to close the gap between the upper bound and lower bound of minimizing general convex finite-sums. For strongly convex finite-sums, the gap is already closed due to (Hannah et al., 2018). We will add the description about the function value complexity of Acc-SVRG-G in the paper.
>
> 2. "The recent work (Lee et al., 2021)".
>
>     Thanks for pointing out this work. We noticed this paper and we are excited about their results, especially the potential function and geometric insight of OGM-G. We feel that stochastic accelerated methods may also have some sort of geometric structure, which could motivate new schemes. We will revise the corresponding parts of our paper regarding their results.
>
> 3. "Theorem 3.2 with a bound on the weighted sum of all iterates does not fully resolve the issue of OGM-G as I hoped from the line 123" / "What do you mean by 'Algorithm 2 converges optimally on the last iterate'?".
>
>     Since Theorem 3.2 implies that $\forall k \in \lbrace 0,\ldots,N \rbrace, ||\nabla f (x_k)||^2 = O(\frac{L\Delta_0}{N^2 \delta_{k+1}})$, each iterate of Algorithm 2 has a bound. And since $\delta_{N+1} = 2$, $x_N$ has the optimal rate in $\epsilon$ (under IFC). We will rewrite Remark 3.2.1 to make these points clearer. We admit that this bound is not the common "guarantee" one would hope for iterative schemes, and thus we will revise Line 123 as "all other iterates of OGM-G are not known to have properly upper-bounded gradient norms".
>
> 4. "The intuition behind the 'warm-up phase'".
>
>     Unfortunately, we don't have any intuition except for it being a dedicated analytic trick. We also don't find any intuitive explanation in [3, 36, 52, 38].
>
>     Technically speaking, denoting the potential function of Acc-SVRG-G as $a_k (f(x_k) - f(x^\star)) + \frac{L}{2} ||z_k - x^\star||^2$, the issue of the single-stage choice is that $a_0 = O(n)$, which leads to the dominant terms $O(\frac{n}{\sqrt{\epsilon}})$ and $O(\frac{n}{\epsilon^{2/3}})$ in the complexities. The two-stage choice aims to ensure that (i) $a_0 = O(1)$, (ii) the potential function is decreasing and (iii) $a_k = \Omega(k^2)$ when $k$ is large.
>
>     This trick does not seem to be necessary in the strongly convex case, in which a constant parameter choice already achieves the optimal complexity.
>
> 5. "The analysis for the low accuracy regime"
>
>     The improvement in the low accuracy regime in Theorem 4.3 is at most $O(\log{n})$ compared with the complexity in Theorem 4.2 (and Katyusha). We mainly want to show that the OGM framework is capable of achieving the best complexities that were derived under the NAG framework. We will add more remarks for Theorem 4.3.
>
> 6. "line 208: Which rate should I compare here?"
>
>     We are comparing with adaptively regularized NAG whose rate is $O(\frac{n}{\sqrt{\epsilon}})$ (versus $O(n+\sqrt{\frac{n}{\epsilon}})$ of R-Acc-SVRG-G, omitting log factors). We will clarify this in the paper.
>
> 7. "The insight of the termination conditions in R-Acc-SVRG-G".
>
>     At outer iteration $t$, if we let the inner loop count $k$->$\infty$, according to Proposition 5.2 (i), $\mathbb{E}[||\nabla f(x_k)||] \leq \delta_t R_0$. If the current estimate $\delta_t$ is large (i.e., larger than $\frac{\epsilon}{R_0}$), it is quite likely that the algorithm will not terminate with $||\nabla f (x_k)|| \leq \epsilon$ at Step 9, even if we run infinite number of inner loops.
>
>     The condition of Step 10 ensures that the algorithm breaks the inner loop when it holds that $\mathbb{E}[||\nabla f(x_k)||] \leq 2\delta_t R_0$. Thus, if the algorithm does not terminate before it breaks at Step 10, it is quite possible that running infinite number of inner loop, the algorithm still will not terminate. We will add more explanation in the paper.
>
> 8. "Sections 3 and 4 are somewhat disconnected".
>
>     Thank you for your comment and we will try to improve the presentation.
>
>     We have made several preliminary attempts to extend OGM-G into the stochastic setting (both finite sum and stochastic approximation). We found that if we directly follow the analysis in Proposition 3.1 for the stochastic variants, the last two summation terms will easily lead to variance accumulation, which destroys the convergence. We feel that using the potential function discovered in (Lee et al., 2021) is a possible solution to fix this issue.
>
>     We also suspect that there exists an accelerated variant of SARAH(L2S) that is built upon OGM-G due to its position in Table 1. The existing analysis of SVRG variants (without regularization) crucially relies on IDC.
>
> 9. "Experiments".
>
>     We admit that the experiments in the current paper are not sufficient to the eyes of practitioners. Currently, we mainly focus on algorithmic techniques and their theoretical guarantees. Conducting extensive experiments is left as an important future work.
>
> For the other comments about uncleared parts or typos at Line 37, 46, 135, 149 and 177, we will revise them according to your suggestions. Thank you very much for pointing them out.
>
> ## Reference
> (Hannah et al., 2018) R. Hannah, Y. Liu, D. O’Connor, and W. Yin (2018). Breaking the span assumption yields fast finite-sum minimization. In NeurIPS, pages 2312–2321.
>
> (Lee et al., 2021) J. Lee, C. Park, E.K. Ryu (2021). A Geometric Structure of Acceleration and Its Role in Making Gradients Small Fast. In arXiv.

---

### Official Review · Reviewer_hnRP · 2021-07-16

**Rating:** 6
**Confidence:** 3

**Summary:**

The paper introduces three new algorithms for making gradient small in (finite-sum) the convex setting: (1) the first algorithm comes with a new parameter setting for OGM-G, matching the lower bound in the deterministic setting. (2) the second algorithm combines loopless-SVRG with OGM framework (3) the third algorithm is an accelerated SVRG with adaptive regularization, matching the lower bound up to logarithmic terms.

**Limitations And Societal Impact:**

Yes.

**Main Review:**

Originality: I believe the ideas and algorithms are original.

Clarity: The paper is well organized and easy to read.

Significance: Finding near-stationary point in convex setting is a critical task and has been explored by many previous work.

Comments
1. My largest concern is the improvement over previous work is incremental. The analysis of OGM-G under a potential function has appears in Diakonikolas and Wang, 2021, although the potential function is not exactly the same. The second algorithm is not optimal in either $n$ or $\epsilon$ in minimizing the gradient norm. The third algorithm does not improve over regularized Katyusha in the complexity. Although it does not require the knowledge of constant such as $R_0$ it still requires to know $L$.

2.  I think one of the drawbacks of algorithm 2 may be that it still needs to prefix the total iteration $N$.

3. Although the second algorithm can be used to both finding small function value and small gradient norm, it outputs different points based on different criterion (last iterate when using optimal gap and a random iterate when using gradient norm). Is it possible to output a point with both small function value and gradient norm?

4. The paper only provides the experiment for algorithm 3 and the comparisons with the state-of-the-art are missing. Since the complexities of algorithms does not improve over the previous algorithms, it benefits to do experiments to compare the performance.

Minor comments

5. In Remark 3.2.1, the last iterates converge follows directly from Theorem 3.2 with $\delta_{N+1} = 2$, which directly implies the minimum gradient norm is small.

6. It would be interesting to explore algorithm without regularization to achieve the lower bound with IDC setting.


**Time Spent Reviewing:**

3

---

> ### Author Response · Authors · 2021-08-08
> **Reply To Reviewer hnRP**
>
> Thank you for taking the time to carefully read the paper and your constructive comments. Your comments make us aware that we haven't adequately emphasized our contributions in the submission, and we were not absolutely careful when estimating the complexity of Algorithm 3. We would like to address each of your concerns as follows.
>
> 1. "Improvement is incremental".
>
>     We clarify the contributions in each section of our paper below (the discussions will also be added to the paper).
> \
> \
>     **(Section 3)** The main contribution of this section is the memory-saving variant M-OGM-G (Algorithm 2), while our potential function analysis does have several advantages over [1]. J. Diakonikolas and P. Wang [1] attempted to analyze and generalize OGM-G. However, as they admitted (at the end of page 10 of [1]), "the entire sequence $\lbrace A_k\rbrace_{k=0}^K$ needs to be pre-computed and stored, which appears to be unavoidable". Thus, M-OGM-G cannot be derived in their analysis. Another work [2] (which appeared online after the submission deadline) also studies variants of OGM-G. Their proposed FISTA-G and FGM-G require a more complicated recursion to be pre-computed and stored. Thus, M-OGM-G does have the benefit of completely dropping this requirement without losing the optimal rate, which fills the void.
>
>     Another interesting part is how M-OGM-G is discovered. The main difficulty of generalizing OGM-G is its extremely tight proof and the lack of insight, which makes it rather hard to find other parameter settings with bare hands. We propose in Section 3.1 a tunable relaxation to the interpolation conditions and use the computer-aided methodology. It is actually the first time computer-aided tools are used in this way. These tools are intrinsically designed to find fastest schemes and give tightest analysis. Our twist on the interpolation conditions allows to identify less tight formulations, which gives room for potential generalization. Moreover, the discovery of M-OGM-G is another example of the powerfulness of computer-aided methodology, which finds proofs that are difficult (or even impossible) to find with bare hands.
>
>     The potential function analysis of OGM-G in Proposition 3.1 has the advantage of being tighter than the one in [1]. The derived rate is $\frac{8L\Delta_0}{(N+2)^2}$ compared with $\frac{16L\Delta_0}{(N+2)^2}$ in [1], and we are capable of showing $\frac{4L\Delta_0}{(N+1)^2}$ if the first step correction is used. Moreover, the proof is simpler than [1] due to the momentum vector $v$, which allows us to naturally expand $x_k - x_N$ at Line 490.
>
>     Last but not least, OGM-G seems to have a completely different acceleration mechanism compared with existing accelerated methods (NAG, HB, OGM, TM...). This attracts several recent efforts devoted to understanding its principle. We believe that we have made a solid step towards this goal.
> \
> \
>     **(Section 4)** We found that we can change the output for gradient in Algorithm 3 (Acc-SVRG-G) as: for all the observed full gradients $\lbrace\lVert \nabla f(x_k)\rVert^2 \rbrace_{k=0}^{K}$, output the point that attains the minimum one. This output has the complexity $O(n\log{\frac{1}{\epsilon} + \min\lbrace\frac{n^{2/3}}{\epsilon^{2/3}}, \frac{\sqrt{n}}{\epsilon}\rbrace})$, which is optimal in $n$ for sufficiently large $n$. This is because we didn't consider the last iterate in the original output.
>
>     We list the features of Acc-SVRG-G below.
>
>     - We use the OGM framework as the inspiration and to distinguish Acc-SVRG-G from the existing accelerated variants. It is clearly the first time OGM is extended to the stochastic case. Note that the original OGM has nothing to do with making the gradient small and there is no hint on how a stochastic variant can be designed. There are infinitely many ways to extend a deterministic method to the stochastic case and it often takes years of trial and error until we finally identify the correct one. For example, the researchers proposed several different direct and indirect accelerated variants of SVRG until they finally discovered that Katyusha is the correct stochastic extension of NAG.
>     - Acc-SVRG-G minimizes the function value at the same rate as ANITA [3], which is the current best-known rate (optimal up to log factors).
>     - Acc-SVRG-G achieves a reasonably good rate for minimizing the gradient norm at the same time, which is clearly the current best rate for a single direct scheme.
>     - Acc-SVRG-G has an arguably simpler structure than its counterparts, which only involves one coupling step.
>     - Acc-SVRG-G has an arguably simpler and tighter proof. No Young's inequality is used.
>
>     Based on these features, we believe that Acc-SVRG-G is worth to be published.
> \
> \
>     **(Section 5)** R-Acc-SVRG-G achieves the near-optimal rates without the knowledge of $R_0$ or $\Delta_0$. Deriving stochastic scheme that adapts to $L$ is another important topic. To the best of our knowledge, there is no truly parameter-free stochastic variance reduced method in the literature. The most recent attempt we are aware of is AI-SARAH [4], while the fully adaptive method in that work is still heuristic. Clearly, all our proposed schemes depend on a known $L$. For tasks such as logistic regression or least squares, $L$ is explicitly known, and there are several existing works about estimating $L$. We feel that the attempt for parameter-free variance reduction does not quite fit the theme of our work. But we also feel that the adaptive regularization framework points out a possible solution.
> \
> \
>     Based on the above points, our work improves the current knowledge of both direct and indirect schemes, and thus we believe that our contributions are substantial.
>
> 1. "Algorithm 2 (M-OGM-G) needs a prefixed $N$".
>
>    It is conjectured in [1] that a prefixed $N$ is necessary for achieving the accelerated rate; otherwise, GD is optimal. OGM-G and M-OGM-G are suitable for tasks where we have a fixed gradient budget. In this case, M-OGM-G has the benefit that we can terminate the algorithm a bit earlier while still enjoy a reasonably good convergence guarantee (according to Theorem 3.2).
>
> 2. "Algorithm 3 outputs different points".
>
>     We are not aware of any existing method that can output a point with both small function value and gradient norm (at the optimal rates). Note that the overhead of different output options is always negligible since the main computational burden is on the gradient evaluation.
>
> 3. "Experiments".
>
>     Conducting extensive experiments to evaluate the proposed schemes is definitely very important, which requires considerable efforts. As mentioned in the abstract, this work is mainly devoted to the study of algorithmic techniques, which is theory-oriented. Nevertheless, we have tried our best to make the proposed schemes as easy to implement as possible, and we also feel that the momentum insight of OGM-G could extend the acceleration effect even beyond the convex setting (as mentioned in Section 6).
>
> 4. "Remark 3.2.1".
>
>    Note that there is no guarantee that the last iterate will always attain the smallest gradient norm. It is because that M-OGM-G is not guaranteed to monotonically decrease the gradient norm (similarly, NAG is not guaranteed to monotonically decrease the function value). The only method that is proved to have this property is GD. Moreover, according to this remark, outputting the smallest gradient actually leads to a better constant factor.
>
> 5. "Explore direct algorithm that achieves the lower bound under IDC".
>
>     In view of the position of SARAH(L2S) in Table 1, we suspect that there exists an accelerated variant of SARAH which obtains the optimal rate under IFC and reduces to OGM-G in the deterministic case. An optimal algorithm under IDC can then be constructed as "Katyusha + Acc-SARAH". We have made several preliminary attempts but all failed due to the limited understanding of OGM-G (regarding its parameter space, acceleration mechanism, etc). Just to point out that even if such a scheme exists, it still cannot minimize the function value and gradient norm at the same time. In this sense, Acc-SVRG-G is not replaceable.
>
>
> Clearly, we are all aiming for the best algorithm, i.e., the one with optimal rate, has a direct scheme, being memory-efficient and parameter-free, which is the ultimate goal of optimization research. However, there are always various technical difficulties that prevent us from achieving the goal. We kindly ask the reviewer to take the above feedbacks into consideration when making the final evaluation of our work.
>
> ## References
> [1] J. Diakonikolas and P. Wang (2021). Potential Function-based Framework for Making the Gradients Small in Convex and Min-Max Optimization. In arXiv.
>
> [2] J. Lee, C. Park, E.K. Ryu (2021). A Geometric Structure of Acceleration and Its Role in Making Gradients Small Fast. In arXiv.
>
> [3] Z. Li (2021). ANITA: An Optimal Loopless Accelerated Variance-Reduced Gradient Method. In arXiv.
>
> [4] Z. Shi, N. Loizou, P. Richtárik, M. Takáč (2021). AI-SARAH: Adaptive and Implicit Stochastic Recursive Gradient Methods. In arXiv.

---

> > ### Comment · Reviewer_hnRP · 2021-08-16
> > **reply**
> >
> > Thank you for your detailed reply! I will reconsider some of my points.

---

### Decision · Program_Chairs · 2021-09-27

**Decision:**

Reject

**Comment:**

The paper considers algorithms to find near stationary points ( $\\| \nabla f(x) \\| \\le \epsilon $ ) for convex finite-sums, as opposed to finding points with small objective value $f(x) - \\min_y f(y) \\le \\epsilon$. They propose three new algorithms.  As one reviewer said during discussion (and seconded by another reviewer), they liked that the proposed M-OGM-G method is produced from a computer-aided method (PEP-style ideas being a new field of optimization), they agree the proof for Acc-SVRG-G is very non-trivial, and also that this problem (of near stationary points) is of interest and has not seen enough attention.

Reviewer 3GpM gave the paper a very high score, but explained more in discussion that this was mainly due to the apparent correctness of the math, and not necessarily to be interpreted as an overall score (in terms of contribution).

The dominant concern of all reviewers (including 3GpM) was that the paper is too incremental. For each of the three algorithms, there are specific improvements, but some of these are slight.  Even the most positive reviewer, 3GpM, was less enthusiastic after reading the author(s) response to reviewer hnRP, as this makes it sound like the improvement is incremental.

Looking at the paper myself, the paper is nicely written, and I like the fact that it reviews recent literature. But it's really a collection of a few small improvements, and I don't think these "sum" up to be equivalent to a major result.  This is nice work, but the end result is just not particularly exciting.  My suggestion is the the authors submit this work, with only minor changes, to a slightly lower tier venue, as I think the work deserves to be published but just not necessarily in a tier-1 venue.